# Association Analysis of Tiller-Related Traits with EST-SSR Markers in *Psathyrostachys juncea*

**DOI:** 10.3390/genes14101970

**Published:** 2023-10-21

**Authors:** Zhen Li, Tian Wang, Lan Yun, Xiaomin Ren, Yong Wang, Fengling Shi

**Affiliations:** 1College of Grassland, Resources and Environment, Inner Mongolia Agricultural University, Hohhot 010010, China; lizhen0471@163.com (Z.L.);; 2Key Laboratory of Grassland Resources Ministry of Education, Hohhot 010010, China

**Keywords:** *Psathyrostachys juncea*, plant architecture, tillering traits, SSR markers, association analysis

## Abstract

*Psathyrostachys juncea* is a long-lived perennial Gramineae grass with dense basal tillers and soft leaves. It is used widely in cold and dry areas of Eurasia and North America to establish grazing pasture and is even used as an ideal plant for revegetation and ecological restoration. Plant architecture, especially tillering traits, is critical for bunch grasses in breeding programs, and these traits in plants are mostly quantitative traits. In this study, the genetic diversity, population structure, and linkage disequilibrium of 480 individual lines were analyzed using 127 pairs of the EST-SSR marker, and a significant association between ten plant-architecture-related traits of *P. juncea* and molecular markers was found. The results of the genetic diversity analysis showed that the number of observed alleles was 1.957, the number of effective alleles was 1.682, Shannon’s information index was 0.554, observed heterozygosity was 0.353, expected heterozygosity was 0.379, and the polymorphism information content was 0.300. A total of 480 individual lines were clustered into five groups based on population genetic structure, principal coordinate analysis, and unweighted pair group method with arithmetic mean analysis (UPGMA). The linkage disequilibrium coefficient (*r*^2^) was between 0.00 and 0.68, with an average of 0.04, which indicated a relatively low level of linkage disequilibrium among loci. The results of the association analysis revealed 55 significant marker–trait associations (MTA). Moreover, nine SSR markers were associated with multiple traits. This study provides tools with promising applications in the molecular selection and breeding of *P. juncea* germplasm.

## 1. Introduction

Russian wildrye grass (*Psathyrostachys juncea* (Fish) Neviski) is a perennial cross-pollinated bunch grass species of Gramineae with a rare diploid chromosome level (2n = 2x = 14) [1]. It is native to central and northern Asia’s inland areas, and its distribution has spread to East Asia, Europe, and North America due to introduction and domestication [2]. In China, Russian wildrye is mainly distributed in the north of the Tianshan Mountains in Xinjiang, which is close to central Asia, and there is scattered distributed in the mid-west of Inner Mongolia and northern Tibet. Russian wildrye is a cool-season forage species that is well adapted to arid and semi-arid climates [3,4]. It is a long-lived perennial bunch grass characterized by dense basal leaves that retain their nutritive value during the late growing season better than other grasses in the North American steppe. Most forage of this species is produced in the basal leaves that grow rapidly in spring and remain palatable throughout the summer and fall as long as soil moisture is available [5]. Russian wildrye has extremely strong drought resistance and cold tolerance; it is adapted to heavy grazing and can restrain weeds once established as cultivated grassland [6]. *P. juncea* is considered an important gramineous forage species with agronomic, economic, forage, and breeding value. Since Russian wildrye is widely used for revegetation and the establishment of grazing pastures, tillering capacity and improvement of plant architecture are current focuses for breeding programs targeting this species.

Many important traits in plants are quantitative traits controlled by micro-effective polygenes, with complex genetic mechanisms and susceptibility to environmental factors [7]. Tillering capacity and plant-architecture-related traits are mostly quantitative traits [8]. Early quantitative trait mapping was mainly based on genetic linkage analysis, linkage mapping, and association mapping [9,10]. With the development of genotyping chips and whole-genome sequencing technology, genome-wide association studies (GWAS) have become increasingly applied to the genetic analysis of various complex traits in plants [11,12]. Association analysis, also known as linkage disequilibrium mapping or association mapping, is an analysis method based on linkage disequilibrium (LD) used to identify the correlation between target traits and genetic markers or candidate genes in a population [13,14,15]. Association analysis is an effective way to study the quantitative traits of plants and has been widely used in crops and other plants. With the continuous development of sequencing technology, a large number of molecular markers have been developed, and association analysis has been widely used in the discovery of genes related to various traits [16,17]. The association analysis method can be used to construct genetic maps and is a fast, low-cost method with a high mapping resolution. It is thus an effective method for discerning the genetic anatomy of complex traits [18]. In plant breeding, association analysis has been employed for detecting genomic regions for various agronomic traits, biochemical properties, root system architecture features, and physiological traits, as well as tolerance to environmental stresses [19,20,21,22]. Simple sequence repeat (SSR) markers are widely used in association analysis due to their large number and high frequency of distribution in plant genomes. SSR markers are economical and efficient when applied to population selection because they are easy to detect with PCR reactions [23,24,25]. Li et al. [26] found four continuous markers of EGA resistance and tolerance in different growth stages of wheat when using SSR markers for association analysis, which provided a basis for molecular assisted breeding of wheat. Pour et al. [27] analyzed the genetic diversity and association of 186 wheat and wild relative germplasms using 24 SSR markers and found that there were eight significant markers associated with APX, CAT, and DM under control conditions and nine significant markers associated with APX, GPX, POX, DM, and STI under drought stress. Breseghello et al. [28] performed association mapping of 95 winter wheat varieties using 88 SSR markers, detected significant markers associated with grain size on three chromosomes, and identified alleles that might be useful for genome selection. Yu et al. [29,30] conducted correlation analysis on the drought and flood tolerance traits of perennial ryegrass using 109 SSR markers, which provided important genetic information for the improvement of ryegrass populations. Nie et al. [31] researched the physiological and photosynthetic characteristics of *Miscanthus* under drought stress and conducted association analysis with SSR molecular markers. Through the GLM model, a total of five markers were found to be associated with photosynthetic traits and two markers were associated with leaf RWC of *Miscanthus*. These associations would serve as an efficient tool for the study of drought response mechanisms in *Miscanthus* and facilitate drought-resistance genetic improvements in this species. Yan et al. [32] used EST-SSR to conduct association analysis on the rust-resistance traits of 75 orchardgrass materials and detected 20 markers associated with rust traits in multiple environments. The 20 markers obtained from association analysis could be used in breeding programs for lineage selection to prevent losses caused by rust and provided valuable information for further association mapping using this collection of orchardgrass. Sun et al. [33] used EST-SSR molecular markers to conduct association analysis on important agronomic traits of *Bromus catharticus* Vahl. A total of 101 loci were detected to be associated with nine traits, of which 14 markers were significantly associated with tiller number and 20 markers were significantly correlated with plant height. That study provided a basis for molecular breeding of *Bromus* L. plants. Previous studies have been reported on the genetic structure and biodiversity of *P. juncea* [34]. It was found that a high level of genetic biodiversity exists between individuals within populations, and this was mainly attributed to its strictly heterogamous pollination characteristics. Individual lines are commonly used as genotyping materials in cross-pollinating plants [35]. However, to date, there have been no relevant reports on plant architecture-related traits of *P. juncea* using association analysis.

Therefore, in this study, association analysis was conducted using a total of 127 pairs of EST-SSR primer pairs, of which 103 primer pairs were developed and screened from transcriptome sequences of *P. juncea*, and 24 primer pairs were selected from NCBI. Four hundred and eighty *P. juncea* individual lines from different countries and regions were used as genotyping and phenotyping materials. The association between EST-SSR molecular markers and ten plant architecture-related traits of *P. juncea* were analyzed. SSR markers significantly associated with critical agronomic traits were detected, which provided potential novel molecular markers for marker-assisted selection and genetic improvement of quantitative agronomic traits of *P. juncea*.

## 2. Materials and Methods

### 2.1. Plant Materials and Tiller Related Traits Phenotyping

A total of 480 *P. juncea* individual lines from 21 accessions with 14 to 30 individuals per accession were selected as test materials. All accessions were provided by the U.S. National Plant Germplasm Resources Conservation System (NPGS) and the China National Medium-term Gene Bank for Forage Germplasm. Information on the accessions and cultivation of experimental plots were described by Li et al. [34].

The 480 individual lines were cloned and transplanted in two plots located in different areas through the isolation of plantlets from the tiller nodes in June 2019. The two test plots were located in Hohhot city (111°50′46.6116′′ N, 40°44′58.3188′′ E) and Baotou city (110°34′42.7512′′ N, 40°35′54.8124′′ E) in Inner Mongolia, China. A randomized block design was used in the experiment. The row spacing was 60 cm, and the plant spacing was 50 cm. Normal weed control and irrigation were undertaken.

Observations and records of tiller-related traits of *P. juncea* were made at the two experimental sites during the annual heading stage from 2020 to 2022. Shoot height (SH), clump basal diameter (BD), canopy diameter (CD), nutritional tiller number (NTN), nutritional tiller angle (NTA), leaf length (LL), leaf width (LW), plant height (PH), reproductive tiller length (RTL) and reproductive tiller number (RTN) were measured once a year in the blooming period. The specific measurement methods have been described in the research conducted by Gao et al. [36].

### 2.2. EST-SSR Markers Development for P. juncea

Two groups of thirty plants with dense and sparse tillers, respectively, were selected from the 480 individual lines. Total RNA was extracted from tiller node tissues using a rapid plant RNA extraction kit. The cDNA library was constructed and sequenced by the Biomarker Biotechnology Corporation (Beijing, China). Briefly, the poly (A) mRNA was enriched via magnetic oligo (dT) beads and then broken into short fragments using an RNA Fragmentation Kit (Beckman Coulter, Brea, CA, USA). These cleaved mRNA fragments were used as templates for cDNA synthesis using random hexamer primers. Next, the purified cDNA fragments were enriched via PCR. Finally, the cDNA library was sequenced using an Illumina HiSeq 2000 (San Diego, CA, USA), and the high-quality sequencing data were assembled to obtain unigenes. Four hundred EST-SSR loci were screened based on significantly differently expressed genes between the dense and sparse tillering groups, and primers of SSRs were designed using Primer 5 software (http://primer5.sourceforge.net/releases.php, accessed on 1 August 2023). After polymorphism testing, a total of 103 pairs of EST-SSR primers with polymorphism were selected. In addition, twenty-four EST-SSR markers related to tillering were selected from NCBI for this experiment. (Appendix A).

### 2.3. Population Genotyping with EST-SSR Markers

The total DNA of 480 individual lines was extracted using a plant DNA extraction kit (Tiangen Biochemical Co., Ltd., Beijing, China), and DNA quality and concentration were tested using 1% agarose gel and a NanoDropTM2000 Spectrophotometer (Thermo Fisher, Waltham, MA, USA). The DNA solutions were diluted to 50 ng/µL. A total of 127 selected EST-SSR primer pairs were used from the 480 individual lines in DNA PCR reactions. The 15 µL PCR reaction system were used for DNA amplification. The PCR reaction and conditions have been described by Nie [31]. The amplified fragments were detected using a QIAxcel Advanced capillary electrophoresis instrument (QIAGEN, Hilden, Germany). The 100 bp to 2.5 kb DNA Size Marker and 15 bp to 3 kb Alignment Marker were used as standard size markers. Amplified products of each individual line and each primer pair were analyzed using BioCalculator software to calculate the peak features. Clear bands with high statistical resolution were counted and arranged in order from small to large, with “1” indicating the presence of a fragment and “0” indicating the absence of a fragment.

### 2.4. Linkage Disequilibrium, Population Structure and Association Analysis

The phenotypic data of ten quantitative traits were processed, and preliminary statistics were conducted using Excel. Correlation analysis and principal component analysis of phenotypic trait data were conducted using R 4.2. For multi-year and multi-location data, the best linear unbiased estimate (BLUE) value was calculated using the R 4.2 lme4 package [37]. The calculation model of the BLUE value was as follows:$$Y_{ikm}=\mu +g_{i}+\tau_{k}+{g\tau }_{ik}+\delta_{(k)m}+\varepsilon_{ikm}$$
where μ is the population average, $g_{I}$ is the genotype effect, $\tau_{k}$ is the environmental effect, ${g\tau }_{ik}$ is the genotype and environment interaction effect, $\delta_{(k)m}$ is the m-th repeating effect in the k-th environment, and $\varepsilon_{ikm}$ is the random error effect. The genotype effect is a fixed effect in the equation, and other factors are random effects.

The genotypic data of EST-SSR markers from 480 individual lines were analyzed, and the number of alleles (Na), the number of effective alleles (Ne), Shannon information index (*I*), observed heterozygosity (*H*_0_), expected heterozygosity (*He*) and polymorphism information content (PIC) were calculated. The analysis model of molecular variance (AMOVA) was used to estimate the genetic variation within and among individual lines using GenAlex 6.3 software. Principal coordinate analysis (PCoA) was also undertaken using GenAlex 6.3 software. Genetic similarity coefficients were calculated, and the un-weighted pair group method with arithmetic mean (UPGMA) dendrogram was constructed using NTSYS-pc 2.10 software. Powermarker V3.25 software was used for cluster analysis. The population genetic structure was analyzed using STRUCTURE 2.3.4 software. To confirm the optimum number of subpopulations (K), ten independent runs for each value of K ranging from 2 to 10 were conducted. Each run consisted of a burn-in period of 10,000 steps followed by 100,000 MCMC iterations. The ΔK parameter, based on the rate of change in the log probability of data between successive K values, was estimated to determine the best K based on the model developed by Evanno (Lausanne, Switzerland) [38].

Linkage disequilibrium between EST-SSR loci was estimated using TASSEL v2.1. software. Association analysis between the ten tiller-related phenotypic traits and EST-SSR molecular markers was performed based on a generalized linear model (GLM) using TASSEL v2.1. Marker-trait association was considered significant using a threshold of *p* < 0.05. The Manhattan diagram was drawn using the CMplot package of R 4.2. 

## 3. Results

### 3.1. Phenotypic Analysis of Tiller-Related Traits

Variation analysis using BLUE values and phenotypic values for ten traits of *P. juncea* in two locations over three years showed that the coefficient of variation ranged from 8.09% to 68.64% among different traits (Table 1). The nutritional tiller angle trait had the lowest coefficient of variation value, which was 8.09%, indicating that the nutritional tiller angle was the most stable trait with minimal variation among all traits evaluated in this experiment. The highest coefficient of variation value was for the reproductive tiller number, which was 68.64% in Hohhot, followed by the nutritional tiller number, which was 40.36% in Hohhot. Analyzing all phenotypic data across both sites and BLUE values, all the coefficients of variation for reproductive tiller number and nutritional tiller number traits exceeded 30%, showing that there was rich variation in both reproductive and nutritional tiller number traits in the *P. juncea* population. Comparing the phenotypic data of the two sites, nine of the ten traits had higher coefficients of variation in Hohhot than in Baotou, suggesting that the environments of the two sites affected these traits in different ways.

Correlation analysis of ten traits of *P. juncea* was carried out with BLUE values. The results showed that plant height, reproductive tiller length and shoot height were significantly positively correlated with all traits (*p* < 0.001). Clump basal diameter was significantly positively correlated with leaf length, leaf width, canopy diameter, nutritional tiller number and reproductive tiller number (*p* < 0.001). Leaf length was significantly positively correlated with all traits (*p* < 0.001). Leaf width was significantly positively correlated with canopy diameter and nutritional tiller number (*p* < 0.001). Canopy diameter was significantly positively correlated with nutritional tiller number and reproductive tiller number (*p* < 0.001). Nutritional tiller angle was significantly positively correlated with reproductive tiller number (*p* < 0.05). The correlation between plant height and reproductive tiller length was the strongest, with a correlation coefficient of 0.96, followed by reproductive tiller length and canopy diameter, with a correlation coefficient of 0.80 (Figure 1A).

Principal component analysis (PCA) was used to evaluate variation among the investigated germplasms. The first four principal components (PCs) accounted for 74.81% (PC1: 46.8%, PC2: 11.1%, PC3: 9.87% and PC4: 7.02) of the total variation, indicating that these four independent comprehensive indicators could basically represent variation among the original ten agronomic traits. The PCA-based biplot showed that all germplasms were widely separated from each other, suggesting a high level of variability among the investigated materials (Figure 1B). In the first principal components corresponding to the eigenvectors, the traits with higher correlation coefficients were plant height, reproductive tiller length, shoot height and clump basal diameter, with correlation coefficients of 0.36, 0.40, 0.37 and 0.31, respectively. These traits mainly reflected the growth status of *P. juncea*. The traits with larger correlation coefficients in the corresponding eigenvectors of the second principal component were leaf length, leaf width and canopy diameter, which mainly reflected the leaves growth of *P. juncea*. The third and fourth principal components corresponded to the eigenvectors of nutritional tiller angle, reproductive tiller number and nutritional tiller number, which mainly reflected important indicators of nutritional and reproductive growth of *P. juncea* (Table 2). Because the first four principal components reflected 74.81% of the information of the original ten single indicators, and considering traits with a greater influence on the growth and tillering ability of *P. juncea*, the four indicators plant height, leaf width, nutritional tiller angle and nutritional tiller number were used for comprehensive evaluation of the growth of *P. juncea*.

### 3.2. The Polymorphism of SSR Markers, Genetic Diversity and Population Structure Analysis

In this study, the genetic diversity of the *P. juncea* population was analyzed using 127 EST-SSR primer pairs, and the results are shown in Appendix A. The number of observed alleles ranged from 1.333 to 2.000, with an average of 1.957. The number of effective alleles ranged from 1.030 to 2.000, with an average of 1.682. The Shannon’s information index ranged from 0.059 to 0.693, with an average of 0.554. The observed heterozygosity ranged from 0.014 to 0.876, with an average of 0.353, and the expected heterozygosity ranged from 0.027 to 0.500, with an average of 0.379. The polymorphism information content ranged from 0.026 to 0.375, with an average of 0.300. In particular, 101 markers had polymorphism information content greater than 0.25, accounting for 79.5% of all markers.

The results of AMOVA revealed that the genetic differentiation coefficient *Fst* was 0.341. Of the 480 individual lines, only 18% of the total genetic variance was due to differences within populations, while 34% was due to differences among populations and 48% was ascribed to differences among individuals within the populations (Table 3).

Principal coordinate analysis (PCoA) provided a better understanding of the relationships among the *P. juncea* population. Results showed that the total variation could be well explained by the first two principal axes, with explanation rates of 15.48% (PC1) and 11.69% (PC2), respectively. The *P. juncea* population of 480 individual lines were clustered into five groups accordingly (Figure 2A).

The Un-weighted Pair Group Method with Arithmetic Mean (UPGMA) dendrogram showed that all 480 *P. juncea* individual lines were clustered into five groups (Figure 2B). Each group contained 96 individuals, accounting for 20% of the total. The results of principal component analysis and cluster analysis were basically consistent. There were significant differences in the sources of *P. juncea* germplasm in each group, and there was no obvious geographic or regional relationship between materials in any group.

The EST-SSR data matrix of 480 individual lines was used for estimating genetic structure. Structure analysis showed that the optimum ∆*K* was 5 (Figure 2C). The 480 individual line populations were clustered into five groups. In group 1, 96 individual plants clustered into one category. In group 2, 192 individual plants clustered into one category. In group 3, 43 individual plants clustered into one category. In group 4, 96 individual plants clustered into one category, and in group 5, 53 individual plants clustered into one category (Figure 2C; Appendix A).

### 3.3. Linkage Disequilibrium and Association Analysis of the P. juncea Population

Across all loci, 5874 locus pairs were detected in *P. juncea* individual lines. Among them, 3914 locus pairs (66.63%) were revealed at the *p* < 0.05 level. The range of D′ was between 0 and 1 (Table 4), and the average value of D′ was 0.3047. The *r*^2^ ranged from 0.00 to 0.68, with an average of 0.04. There were 650 combinations with *r*^2^ ≥ 0.1, accounting for 11.07% of the total, indicating that the LD level of *P. juncea* germplasms was relatively low (Figure 3).

The association analysis between ten tiller-related traits and EST-SSR marker data was computed based on the GLM method. At the *p* < 0.05 level, a total of 55 SSR loci were detected as being associated with ten traits, such as plant height (3), reproductive tiller length (4), shoot height (4), clump basal diameter (4), leaf length (4), leaf width (6), canopy diameter (4), nutritional tiller number (6), nutritional tiller angle (3) and reproductive tiller number (17). The proportion of variation in associated markers explained ranged from 1.060% to 9.210%, with the lowest for marker 000516 associated with reproductive tiller number and the largest for marker *OsIPT7* associated with leaf length. Reproductive tiller number was significantly associated with the most numerous loci (i.e., 17 loci), but they all explained a low proportion of variation. This may suggest that the formation of reproductive tillers is regulated by multiple minor genes and is influenced by the environment; hence, the proportion of variation in reproductive tiller number explained by each locus was relatively low. In addition, 9 out of 55 loci were detected as being associated with multiple traits (Table 5; Figure 4). Marker 139768A is associated with nine of ten traits, excluding reproductive tiller number. Marker *OsIPT7* is associated with eight of ten traits, excluding clump basal diameter and reproductive tiller number. Marker *OsCKX1* is associated with five traits, including reproductive tiller length, shoot height, leaf length, leaf width and canopy diameter. Marker *OsYUCCA6* is associated with five traits, including shoot height, leaf length, leaf width, nutritional tiller angle and canopy diameter. Marker 028324 is associated with four traits, including plant height, reproductive tiller length, clump basal diameter and reproductive tiller number. Markers 019368C and 070047 are both associated with two traits: clump basal diameter and reproductive tiller number. Marker 061356 is associated with two traits: leaf width and nutritional tiller number. Marker 185875 is associated with two traits: nutritional tiller number and reproductive tiller number.

## 4. Discussion

### 4.1. Phenotyping Analysis of the P. juncea Population

Tillering capacity and plant architecture-related traits are critical for bunch grass improvement, whether as a forage or for vegetation restoration. *P. juncea* is a cross-pollinated perennial plant. The ten phenotypic traits investigated in this study showed great variation, especially reproductive tiller number and nutritional tiller number, and revealed a complex genetic background for the quantitative traits of this cross-pollinated species.

Simple statistical analysis of ten phenotypic traits related to tillering of *P. juncea* showed that the coefficient of variation of the reproductive tiller number was the most obvious, followed by the nutritional tiller number (Table 1), which suggested a relatively complex regulating mechanism of tillering ability in this bunch grass. The germplasm materials selected in this study were from eight countries and regions, including China, Mongolia, Russia and the United States. There was a large genetic distance between germplasms, and the related genes varied greatly in the population.

Correlation analysis showed that plant height and reproductive tiller length were significantly positively correlated with all traits (*p* < 0.01). Plant height is an important agronomic trait of forage plants that can represent growth and development ability and can also reflect the biomass yield potential of forage plants [39]. Plant height is also significantly related to seed yield in crop plants, and appropriate plant height can increase the seed yield of wheat [40].

Several reports have indicated that PCA is an efficient multivariate method to display the level of phenotypic variation and identify associations between measured traits. PCA is a multivariate statistical method that transforms multiple indicators into several comprehensive indicators with little loss of information [41]. In this study, PCA showed that four components captured 74.81% of the total variation among the ten traits (Figure 1B). Further analysis of the relationship between trait vectors showed that the tillering angle had no correlation with clump basal diameter, leaf width or canopy diameter, which was consistent with the correlation analysis results. Further analysis of the first four principal components revealed that the four independent comprehensive indicators could represent the original ten agronomic traits. The first principal component mainly reflected the growth height of each part of the plant, the second principal component represented the growth of plant leaves, and the third and fourth principal components mainly reflected the plant’s tillering ability. Among the ten traits, plant height, leaf width, nutritional tiller angle and nutritional tiller number could be used to comprehensively evaluate the overall growth of *P. juncea*.

### 4.2. Genetic Diversity Analysis of P. juncea Material

Genetic diversity analysis is widely used in the evaluation of biodiversity, mainly to reveal biological genetic information and the genetic variation between different germplasms [42]. Plants will mutate in the process of adapting to environmental changes. The better the adaptability of plants to the environment, the greater the degree of variation and the higher the genetic diversity [43]. With the development of molecular marker technology, genetic diversity analysis has been widely used in *Triticum aestivum* [44,45], *Lolium perenne* [46,47], *Medicago sativa* [48] and other plants. Pour-Aboughadareh et al. [49] used the ISSR technique to reveal high genetic diversity among *Triticum boeoticum* populations collected from different regions of Iran. Nie et al. [50] used SSR markers to analyze the genetic diversity of ryegrass, and the results showed that the ryegrass germplasm had high genetic diversity. In this study, the polymorphism content of 79.5% of primers was greater than 0.25, indicating a high level of primer polymorphism. The Shannon’s information index can also be used as a parameter to evaluate genetic diversity. In this study, the average value of I was 0.554, which was derived from the analysis of 127 SSR primer pairs, and was slightly higher than that of 0.420 quantified in a *P. juncea* population using 103 EST-SSR primer pairs by Li et al. [34]. In summary, the primers selected in this study reasonably reflected the genetic diversity of *P. juncea*, and indicated a relatively high level of genetic diversity in the *P. juncea* population. Genetic variation within the individuals (48%) was higher than that among populations (34%) in the present study. *P. juncea* is a cross-pollinated plant species, and a high level of gene exchange between plant individuals contributes to the high level of genetic variation.

### 4.3. Population Structure of P. juncea Germplasm Resources

Population structure analysis is an effective way to reveal the genetic characteristics of germplasm and is a necessary prerequisite for association analysis that can effectively eliminate false associations caused by false positives in the association process [13]. The *P. juncea* germplasm in the present study was clustered into five groups based on PCoA, UPGMA, and STRUCTURE analyses. The classification groups of the 480 *P. juncea* individual materials indicated via PCoA and UPGMA were basically consistent with the analysis of population genetic structure, but some individual materials were clustered into different groups when classified using different methods. The results of the three methods showed that the classification of *P. juncea* with simpler genetic structures was basically consistent, while those with more complex genetic structures were classified into different groups. This difference indicates *P. juncea* high breeding and hybridization potential. The division of groups did not show obvious geographical relationships among *P. juncea* germplasm. This may be due to the cross-pollination and self-incompatibility of *P. juncea*, resulting in continuous gene exchange between materials from different regions during the breeding process, suggesting that *P. juncea* gradually shows a more consistent genetic differentiation, which is similar to the results of Yu et al. [29].

### 4.4. Linkage Disequilibrium Analysis of P. juncea Population

Linkage disequilibrium analysis is the basis of association analysis. The size of the LD value has an influence on the accuracy of the correlation analysis. Therefore, LD analysis is a fundamental step in association analysis. The degree of LD between loci can be reflected in the D′ value and *r*^2^ [51,52]. Qi et al. [20] used SSR markers to analyze the linkage disequilibrium of barley populations and found that the range of D′ values was mainly concentrated between 0 and 0.6, accounting for 82.06% of the paired 992 SSR marker sites. Wang et al. [53] used 262 SSR markers to analyze rice populations and found that 5305 pairs showed significant LD (based on D′, *p* < 0.05), accounting for 15.52% of all pairs of loci, with a D′ value of 0.76. In this study, it was found that D′ was mainly concentrated in the range 0–0.4, accounting for 74.29% of all pairs of loci. The linkage disequilibrium coefficient (*r*^2^) was between 0.00 and 0.68, with an average of 0.04. The level of linkage disequilibrium between loci was lower than those of barley and rice, which may be because the germplasm affected the LD value by constantly changing the outcrossing rate in the process of breeding and selection. It may also be due to the wide geographic range of the source of *P. juncea* germplasm and the population’s high genetic diversity level, so the germplasm population contained a variety of different genes, resulting in a low LD level [54].

### 4.5. Association Analysis of P. juncea

Association analysis can link plant genotypes with phenotypes, and the results of population structure can be used as covariates for association analysis, which can improve the credibility of association analysis. Therefore, the Q value was used as a covariate to control false positive pseudo-association in the association analysis, and the contribution rate can be used to express the degree of influence of the associated loci on the traits [55]. Tillering capacity and plant architecture-related traits are quantitative traits. In the process of plant breeding, certain breeding purposes can be achieved by altering quantitative traits [56,57]. However, quantitative traits are complex and controlled by multiple genes, added to which there are different degrees of variation among genes, which makes the research more difficult [58,59]. Therefore, it is essential to use association analysis to clarify the correlation between genotype and phenotype.

After detecting a high level of genetic and molecular variability in the studied germplasm, we used a GLM model-based marker–trait association analysis (MTA) to incorporate phenotypic and genotypic data [60]. The results of the association analysis indicated that the GLM model was effective in detecting significant MTAs in the perennial grass *P. juncea*. This finding is in agreement with previous reports showing a high efficiency of the GLM method to identify significant associations in crop plants [61,62]. To date, association analysis has been widely used in plant breeding for barley [63], soybean [64] and cotton [65], but there have been no reports of association analysis for *P. juncea*.

In this study, a total of 55 markers were detected as being associated with ten agronomic traits, with a contribution rate of 1.060%~9.210%. The marker with the highest contribution was *OsIPT7*, with a contribution rate of 7.07–9.21%. Further analysis revealed that one locus was associated with multiple traits and one trait was associated with multiple loci. Referring to the existing literature, we noticed that Lai [66] conducted association analysis on 55 barley varieties and found that five traits, including plant height, spike length, stem length, thousand-grain weight, and whole growth period, were associated with 4, 6, 4, 6 and 2 SSR loci, respectively. In our study, each trait was associated with 3–17 markers, among which reproductive tiller number was associated with the largest number of loci (i.e., 17). However, the overall contribution rates of each locus were relatively low, and the differences among them were small. We speculate that this may be due to the fact that the reproductive tiller number trait is controlled by multiple minor genes, so the contribution rates of each locus to the variation in the number of reproductive tillers are relatively low [67]. In this experiment, a total of nine loci associated with multiple traits were found, which were markers 028324 (4), 139768A (9), *OsCKX1* (5), *OsIPT7* (8), 185875 (2), *OsYUCCA6* (5), 019368C (2), 070047 (2) and 061356 (2). Correlation analysis of ten different traits found that the correlation between many traits reached a statistically high or extremely high significance level. At the same time, the principal component analysis converted the ten traits into four comprehensive indicators for a detailed evaluation of *P. juncea* germplasm. Therefore, we speculate that this phenomenon may be due to the genetic correlation of these traits in inheritance. Further analysis of nine multiple traits-associated loci, found that SSR markers amplified from primers designed from the *Oryza sativa* genome, *OsCKX1*, *OsIPT7*, and *OsYUCCA6* were associated with more traits in *P. juncea*. These loci have been reported to be related to tillering and growth ability in rice. These results suggested the high efficiency of using genetically close model species genes to develop primers for non-model plant species plant growth, development, and tillering traits [68,69,70]. In our previous research, we also screened a large number of genes related to the growth and development of *P. juncea*, including *CKX*, *IPT*, and *YUCCA* genes. That research found that *YUCCA* is a key gene in the IAA biosynthesis pathway controlling the growth of tillering buds in *P. juncea*. We also found that *IPT* and *CKX* can affect cytokinin content in *P. juncea*. CTK is mainly synthesized in the root and transported upward in the xylem through transpiration, which can relieve the apical dominance caused by IAA and affect the growth and development of tillers and the number of tillers by promoting the germination and elongation of tiller buds [71,72]. Therefore, the overexpression of *IPT* and the low expression of *CKX* are clear ways to increase CTK levels and yields by increasing the tiller number in *P. juncea* [73]. In addition, we cloned the three genes and analyzed the expression of the genes and protein function. The results showed that *YUCCA* is a down-regulated gene for tillering in *P. juncea*, and its subcellular localization was in the cytoplasm [74]. *IPT* is an up-regulated gene that regulates tillering in *P. juncea*, and its subcellular localization was in the chloroplasts [75]. *CKX* gene plays a down-regulated role in regulating the tillering of *P. juncea*, located subcellular within the cell [76]. Transient transfection and Western blot of the three genes showed that the proteins could be expressed normally. The phylogenetic tree showed that *P. juncea* had the closest relationship with *Triticum aestivum* L. and other wheat crops. These results can provide a theoretical basis for the verification of gene function and precise improvement of traits in *P. juncea*. In this study, *OsCKX1*, *OsIPT7*, and *OsYUCCA6* also showed strong associations, which provides a theoretical reference for genetic improvement and molecular marker-assisted breeding of *P. juncea* germplasm resources in the future. At the same time, we also found that the marker 028324 and the marker 139768A were associated with multiple traits. Although their contribution rates were low, statistical significance was relatively high, indicating a significant correlation between the locus and the traits. This result has great significance for the subsequent development of *P. juncea*-specific trait markers. Surprisingly, in our study, it was found that plant height and reproductive tiller length were highly significantly correlated with all traits (*p* < 0.01). In association analysis, the loci associated with plant height and reproductive tiller length were also associated with multiple other traits. Therefore, selecting traits with strong correlations may also be a consideration when assessing associations between a given locus and multiple traits, as was also pointed out by Jia [77] in his research on *Elymus sibiricus* L.

## 5. Conclusions

In this study, phenotypic analysis was undertaken of ten tiller-related traits of 480 *P. juncea* germplasms. We found that there was significant genetic variability, resulting in rich phenotypic variation in ten traits of *P. juncea*. Further association analysis was conducted on the ten phenotypic traits, and EST-SSR molecular markers of *P. juncea* and 55 SSR loci related to tillering and architecture-related traits were screened. These results provided basic data and a theoretical reference for genetic improvement and molecular marker-assisted breeding of *P. juncea* and other perennial grass species.

## Figures and Tables

**Figure 1 genes-14-01970-f001:**
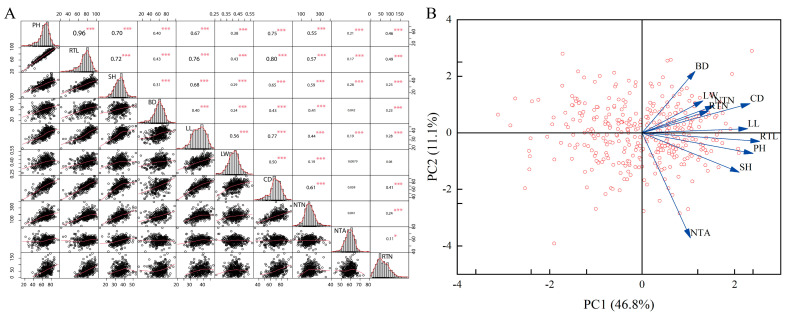
Pearson correlation coefficients and PCA for the ten traits evaluated in the two locations. Statistical significance of coefficients labeled as *** for *p* < 0.0001 and * for *p* <0.05, respectively. Note: (**A**) Pearson correlation coefficients. (**B**) PCA of 480 accessions.

**Figure 2 genes-14-01970-f002:**
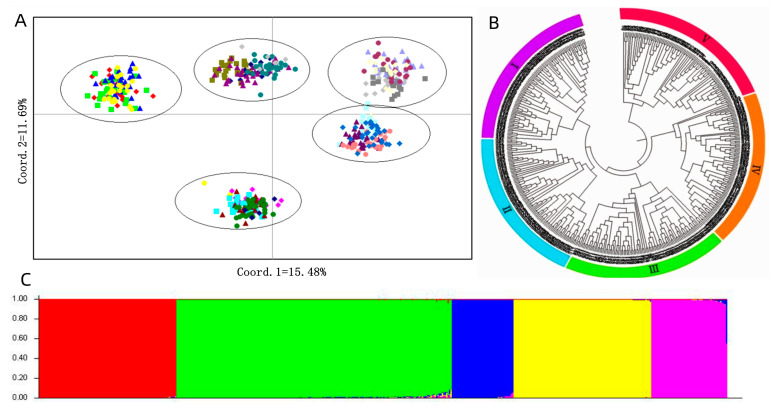
PCA, UPGMA tree and Population Structure of *P. juncea*. (**A**) Principal component analysis (PCA) of 480 accessions. (**B**) UPGMA tree. (**C**) Population Structure. Note: (**A**) Different colors represent different germplasm of *P. juncea.* (**B**,**C**) different colors represent different groups.

**Figure 3 genes-14-01970-f003:**
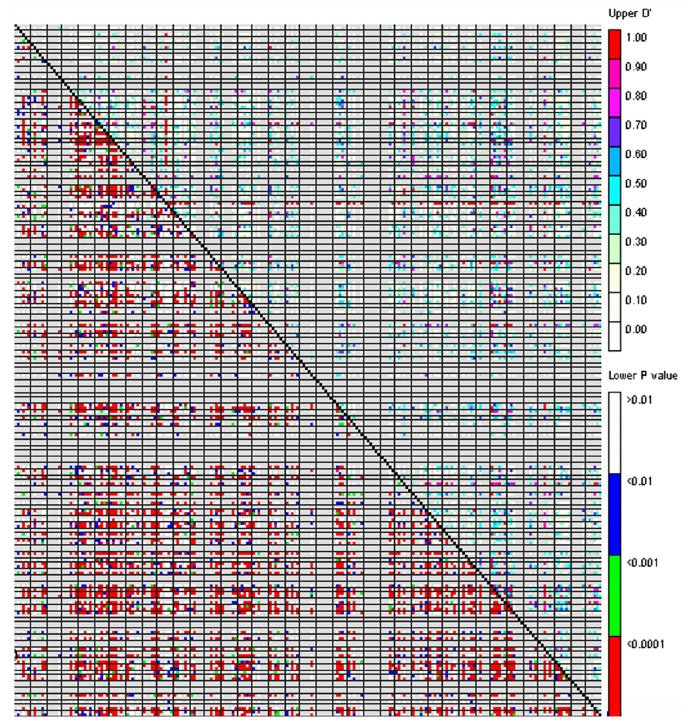
Linkage disequilibrium (LD) between SSR markers.

**Figure 4 genes-14-01970-f004:**
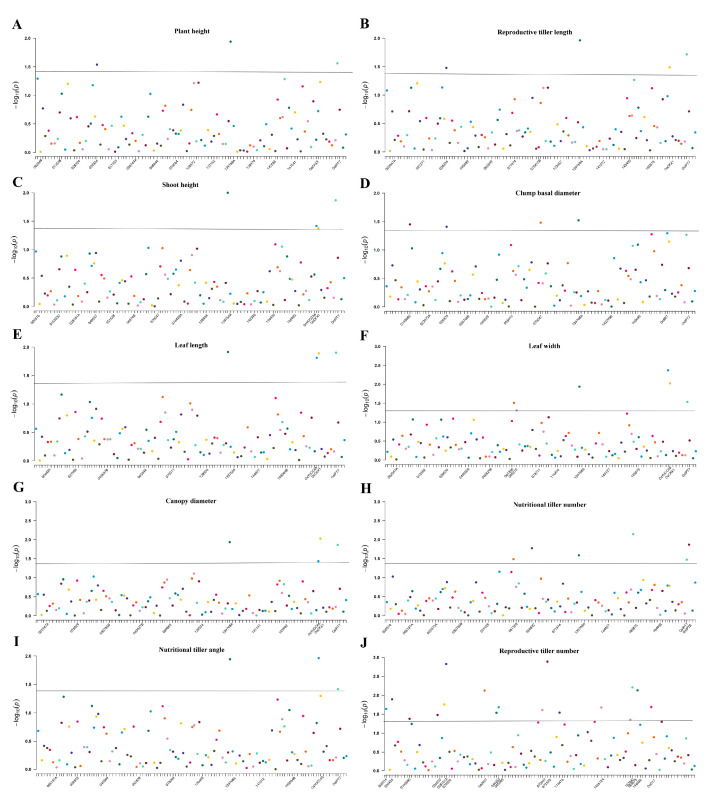
Manhattan plot of SSR markers associated with traits of *P. juncea.* Note: (**A**) PH, (**B**) RTL, (**C**) SH, (**D**) BD, (**E**) LL, (**F**) LW, (**G**) CD, (**H**) NTN, (**I**) NTA, (**J**) RTN.

**Table 1 genes-14-01970-t001:** Descriptive analysis of ten agronomic traits of *P. juncea* population in two environments and their BLUE value.

Trait	Environment	Range	Mean	SD	Coefficient of Variation (%)
Plant height	Hohhot	19.00~91.73	60.91	14.82	24.33
	Baotou	16.00~99.67	70.51	12.33	17.49
	BLUE	14.44~95.13	65.21	11.74	18.01
Reproductive tiller length	Hohhot	23.17~107.43	72.19	14.77	20.46
	Baotou	25.00~111.08	82.90	13.56	16.36
	BLUE	22.16~100.77	76.92	12.61	16.39
Shoot height	Hohhot	12.60~60.90	30.88	8.07	26.15
	Baotou	12.75~61.17	39.32	6.69	17.03
	BLUE	12.52~53.10	34.84	5.50	15.79
Clump basal diameter	Hohhot	15.53~88.13	55.16	11.56	20.96
	Baotou	15.50~108.67	66.38	17.65	26.59
	BLUE	11.09~97.06	60.23	12.80	21.25
Leaf length	Hohhot	15.56~50.05	33.56	5.50	16.39
	Baotou	20.33~52.73	40.46	5.55	13.72
	BLUE	20.08~47.47	36.86	4.59	12.45
Leaf width	Hohhot	0.24~0.60	0.42	0.06	13.98
	Baotou	0.23~0.60	0.41	0.05	12.93
	BLUE	0.26~0.56	0.42	0.05	11.04
Canopy diameter	Hohhot	25.53~102.75	66.79	12.55	18.78
	Baotou	31.50~105.83	74.29	12.36	16.64
	BLUE	33.34~96.02	70.12	10.44	14.89
Nutritional tiller number	Hohhot	17.17~453.67	152.36	61.49	40.36
	Baotou	18.00~426.00	202.19	69.92	34.58
	BLUE	15.36~397.67	174.90	53.51	30.60
Nutritional tiller angle	Hohhot	27.50~83.17	56.44	9.16	16.23
	Baotou	39.17~80.00	61.03	5.64	9.25
	BLUE	41.07~79.16	58.73	4.75	8.09
Reproductive tiller number	Hohhot	4.00~182.00	45.46	31.20	68.64
	Baotou	8.00~325.00	89.25	45.23	50.68
	BLUE	3.10~186.18	59.90	32.31	53.95

**Table 2 genes-14-01970-t002:** Principal component analysis of agronomic traits of *P. juncea*.

Traits	Indicator Eigenvector			
	I	II	III	IV
Plant height	0.40 *	−0.16	−0.10	−0.29
Reproductive tiller length	0.36 *	−0.06	−0.07	−0.16
Shoot height	0.37 *	−0.27	0.28	0.34
Clump basal diameter	0.31 *	0.07	−0.21	−0.01
Leaf length	0.28	0.30 *	0.16	0.01
Leaf width	0.13	0.72 *	−0.07	−0.06
Canopy diameter	0.30	0.23 *	0.02	−0.05
Nutritional tiller number	0.25	−0.09	0.34	0.79 *
Nutritional tiller angle	0.12	−0.41	0.75 *	−0.24
Reproductive tiller number	0.27	−0.22	0.40 *	−0.29
Contribution (%)	46.82	11.10	9.87	7.02
Cumulative Contribution (%)	46.82	57.92	67.79	74.81

Note: The PC coefficient significantly correlated with each trait have been indicated by “*”.

**Table 3 genes-14-01970-t003:** Analysis of molecular variance (AMOVA) of *P. juncea* population.

Sources of Variation	Degrees of Freedom	Sum of Squares	Mean Square	Variance of Components	Percentage Variation (%)	*p* Value	*Fst*
Among Pops	20	18,235.035	911.752	18.957	34	<0.001	0.341
Among Individuals	459	21,331.484	46.474	9.826	18	<0.001	
Within Individuals	480	12,874.500	26.822	26.822	48	<0.001	
Total	959	52,441.020		55.605	100	<0.001	

**Table 4 genes-14-01970-t004:** Number of LD pairs in different D′ value range.

Range of D′ Value	No. of Loci Pairs with Significant LD	Percentage of Loci Pairs with Significant LD (%)
0~0.2	2292	39.02
0.2~0.4	2072	35.27
0.4~0.6	886	15.08
0.6~0.8	342	5.82
0.8~1	282	4.80

**Table 5 genes-14-01970-t005:** EST-SSR markers associated with tiller related traits of *P. juncea*.

Traits	Markers	*p* Value	*R*^2^/%	Traits	Markers	*p* Value	*R*^2^/%
Plant height	028324	0.029	2.290	Nutritional tiller number	061356	0.033	2.230
	139768A	0.012	1.910		066842	0.017	2.530
	*OsIPT7*	0.028	7.840		139768A	0.026	1.520
Reproductive tiller length	028324	0.033	2.190		185875	0.007	2.500
	139768A	0.011	1.930		*OsIPT7*	0.034	7.400
	*OsCKX1*	0.033	1.490		*OsIPT8*	0.014	1.960
	*OsIPT7*	0.019	8.500	Reproductive tiller number	000516	0.023	1.060
Shoot height	139768A	0.010	1.940		004435	0.013	1.880
	*OsYUCCA6*	0.039	1.430		019368C	0.042	1.580
	*OsCKX1*	0.042	1.360		026620	0.033	1.950
	*OsIPT7*	0.014	9.150		028161A	0.018	1.960
Clump basal diameter	019368C	0.036	1.680		028324	0.002	4.130
	028324	0.039	2.060		040957	0.008	4.190
	070047	0.033	3.200		044262	0.029	3.350
	139768A	0.030	1.500		045589	0.021	1.100
Leaf length	139768A	0.012	1.870		070047	0.025	1.630
	*OsYUCCA6*	0.016	1.840		072525	0.001	2.100
	*OsCKX1*	0.013	1.880		114416	0.029	1.550
	*OsIPT7*	0.013	9.210		142612A	0.021	1.600
Leaf width	061356	0.031	2.390		167041	0.044	2.550
	065010	0.049	2.530		185875	0.006	2.670
	139768A	0.012	1.910		194938	0.007	2.620
	*OsYUCCA6*	0.004	2.420		*OsD17*	0.020	1.840
	*OsCKX1*	0.010	2.030	Canopy diameter	139768A	0.012	1.910
	*OsIPT7*	0.029	7.650		*OsYUCCA6*	0.037	1.470
Nutritional tiller angle	139768A	0.011	1.890		*OsCKX1*	0.009	2.030
	*OsYUCCA6*	0.011	1.990		*OsIPT7*	0.014	9.100
	*OsIPT7*	0.038	7.070				

Note: *R*^2^ represents the explanatory rate of phenotypic variation. *IPT* (Isopentenyl transferases); *CKX* (Cytokinin oxidase/dehydrogenase); *YUCCA* (Indole-3-pyruvate monooxygenase); *D17* (Dwarf 17).

## Data Availability

The raw sequencing reads in this study are available in the Sequence Read Archive (SRA) database (accession number PRJNA789128).

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
