# Peer review of "Association Analysis of Tiller-Related Traits with EST-SSR Markers in Psathyrostachys juncea"

_genes, 2023, doi:10.3390/genes14101970_

Round 1

Reviewer 1 Report

1. Improve the introduction section with recent studies, and write the major scope of this study. 

2. Write the plant growth stages and references for measuring the agronomic traits

3. Add the sequencing data NCBI accession number

4. Which statistical model was used to confirm the marker-trait association 

5. Two locations over three years; the statement is confusing in the results section. Whether three years authors planted two locations or just two environment data, make it clear

6. The observed allele number ranged from 1.333 to 2.000, with an average of 1.957. The effective alleles number ranged from 1.030 to 2.000, with an average of 1.682. It is raising concern, about whether the population is diversed and whether the number of markers screened was enough?

7. In table 5;  Please write the footnotes for abbreviation. Mainly include the details for marker ID. Whether R2/% is a phenotypic variance? it is very low and not enough to consider as MTA.  

Moderate editing of the English language required

Reviewer 2 Report

Manuscript "Association Analysis of Tiller-related Traits with EST-SSR Markers in Psathyrostachys juncea" is very intersting.

Authors analyzed the genetic diversity, population structure and linkage disequilibrium of 480 individual line by 127 pairs of EST-SSR marker.
Authors analyzed significant association between ten plant architecture related traits of P. juncea and molecular markers.

The manuscript was written very well. The introduction and description of the methodology are excellent. The results are presented well. The discussion of the results obtained is also interesting.

Minor comments.
Table 1: provide labels for homogeneous groups for comparison of mean values.
Table 2: indicate which PC coefficients were significantly correlated with each trait.
Figure 2A: give the percentages of variation explained by PC1 and PC2.
Figure 4: With a logarithm with base '10', the number '10' is not written. It should be corrected.

Paper needs monor revision.

Round 2

Reviewer 1 Report

I recommend the manuscript for publication to Genes. 

Minor editing of the English language required